

# Follow-up focused on psychological intervention initiated after intensive care unit in adult patients and informal caregivers: a systematic review and meta-analysis

Shodai Yoshihiro[1], Shunsuke Taito[2,3], Kota Yamauchi[4], Shunsuke Kina[5], Takero Terayama[6], Yusuke Tsutsumi[3,7,8], Yuki Kataoka[3,9,10,11] and Takeshi Unoki[12]

[1] Department of Pharmaceutical Services, Hiroshima University Hospital, Hiroshima, Japan
[2] Division of Rehabilitation, Department of Clinical Practice and Support, Hiroshima University Hospital, Hiroshima, Japan
[3] Scientific Research WorkS Peer Support Group (SRWS-PSG), Osaka, Japan
[4] Division of Rehabilitation, Steel Memorial Yawata Hospital, Fukuoka, Japan
[5] Division of Rehabilitation, Nakagami Hospital, Okinawa, Japan
[6] Department of Psychiatry, School of Medicine, National Defense Medical College, Tokorozawa, Japan
[7] Department of Emergency Medicine, National Hospital Organization Mito Medical Center, Ibaraki, Japan
[8] Department of Human Health Science, Graduate School of Medicine, Kyoto University, Kyoto, Japan
[9] Department of Internal Medicine, Kyoto Min-iren Asukai Hospital, Kyoto, Japan
[10] Section of Clinical Epidemiology Section, Department of Community Medicine, Kyoto University Graduate School of Medicine, Kyoto, Japan
[11] Department of Healthcare Epidemiology, Graduate School of Medicine and Public Health, Kyoto University, Kyoto, Japan
[12] Department of Acute and Critical Care Nursing, School of Nursing, Sapporo City University, Sapporo, Japan

Corresponding author
Shunsuke Taito,
shutaitou@hiroshima-u.ac.jp

## ABSTRACT

Psychological dysfunction is one of the considerable health-related outcomes among critically-ill patients and their informal caregivers. Follow-up of intensive care unit (ICU) survivors has been conducted in a variety of different ways, with different timing after discharge, targets of interest (physical, psychological, social) and measures used. Of diverse ICU follow-up, the effects of follow-ups which focused on psychological interventions are unknown. Our research question was whether follow-up with patients and their informal caregivers after ICU discharge improved mental health compared to usual care. We published a protocol for this systematic review and meta-analysis in https://www.protocols.io/ (https://dx.doi.org/10.17504/protocols.io.bvjwn4pe). We searched PubMed, Cochrane Library, EMBASE, CINAHL and PsycInfo from their inception to May 2022. We included randomized controlled trials for follow-ups after ICU discharge and focused on psychological intervention for critically ill adult patients and their informal caregivers. We synthesized primary outcomes, including depression, post-traumatic stress disorder (PTSD), and adverse events using the random-effects method. We used the Grading of Recommendations Assessment, Development and Evaluation approach to rate the certainty of evidence. From the 10,471 records, we identified 13 studies ($n = 3,366$) focusing on patients and four ($n = 538$) focusing on informal caregivers. ICU follow-up for patients resulted in little to no difference in the

prevalence of depression (RR 0.89, 95% CI [0.59–1.34]; low-certainty evidence) and PTSD (RR 0.84, 95% CI [0.55–1.30]; low-certainty evidence) among patients; however, it increased the prevalence of depression (RR 1.58 95% CI [1.01–2.46]; very low-certainty evidence), PTSD (RR 1.36, 95% CI [0.91–2.03]; very low-certainty evidence) among informal caregivers. The evidence for the effect of ICU follow-up on adverse events among patients was insufficient. Eligible studies for informal caregivers did not define any adverse event. The effect of follow-ups after ICU discharge that focused on psychological intervention should be uncertain.

## INTRODUCTION

Adult patients who are admitted to intensive care units (ICU) and their informal caregivers may experience psychological dysfunction, which can persist following discharge (*Needham et al., 2012*). Psychological dysfunction of critically-ill adult patients and their informal caregivers is called post intensive care syndrome (PICS) and PICS-Family (PICS-F), respectively. Other symptoms of PICS include cognitive and physical impairments. Previous studies found that the prevalence of these patients with depression, post-traumatic stress disorder (PTSD), and anxiety was approximately 29% (*Rabiee et al., 2016*), 34% (*Parker et al., 2015*), and 34% (*Nikayin et al., 2016*) after one year of ICU discharge. Studies have also reported that the prevalence of acquired psychological dysfunction among informal caregivers was similar to that among patients (*Johnson et al., 2019*). Therefore, psychological dysfunction is a considerable health-related outcome among critically-ill patients and their informal caregivers.

According to the current guidelines and a systematic review (SR), follow-up with patients who have been admitted to the ICU is comprised of a variety of contents, targets, and times of initiation (*National Institute for Health and Care Excellence, 2009*; *Rosa et al., 2019*). The National Institute for Health and Clinical Excellence guidelines for follow-ups recommended providing enhanced or individualized physical intervention from early mobilization to home rehabilitation (*National Institute for Health and Care Excellence, 2009*). One SR found that the intervention that was initiated in the ICU and continued after ICU discharge included diary and physical rehabilitation (*Rosa et al., 2019*). In addition, the SR did not separately investigate patients and informal caregivers. Similarly, the counterplan for PICS-F was the ICU diary and communication in the ICU. Another SR showed that care providers and informal caregivers regarded the ICU diary as beneficial (*Brandao Barreto et al., 2021*), while another SR asserted that communication in the ICU might reduce symptoms of depression and PTSD (*DeForge et al., 2022*). It would be obvious that these interventions which initiated in the ICU reduced psychological problems of patients and informal caregivers. Moreover, a recent SR studied psychological intervention for patients' informal caregivers, but did not separately investigate adult patients and
pediatric patients (*Cherak et al., 2021*). In a pediatric randomized controlled trial (RCT), interventions were specifically designed for children such as skin-to-skin contact (*Mörelius et al., 2015*), kangaroo care (*Ettenberger et al., 2017*), or guidance for baby care (*Fotiou et al., 2016*). There was clinical heterogeneity among the included studies in the previous SR. Hence, the effects of follow-ups for adult patients and informal caregivers that focused on psychological interventions after ICU discharge have remained unknown.

Thus, the objective of this systematic review and meta-analysis (SR/MA) was to investigate the following research question: does follow-up with adult patients and their informal caregivers following ICU discharge improve mental health compared to usual care?

## MATERIALS & METHODS

### Protocol and registration

We published a protocol for this SR/MA in http://www.protocols.io (*Yoshihiro et al., 2021*). We conducted this SR/MA in accordance with guidelines prescribed by the Cochrane Handbook for Systematic Reviews of Interventions (*Higgins et al., 2020*) and Preferred Reporting Items for Systematic Reviews and Meta-Analysis (PRISMA) (*Page et al., 2021*). The principles listed in the PRISMA statement formed the basis of our SR/MA report (*Page et al., 2021*) (Table S1).

### Eligibility criteria
#### Studies

We included randomized controlled trials that assessed the effects of follow-up after ICU discharge on mental health outcomes among adult patients and informal caregivers. We analyzed papers including published and unpublished articles, abstracts of conferences, and condolence letters. We excluded studies with cluster randomized or quasi-randomized trials, cohort studies, case-control studies, and case series. Furthermore, while including studies for this SR/MA, we did not apply restrictions pertaining to language, country, observation period, or publication year.

In May 2021, we searched the following databases: MEDLINE (PubMed), the Cochrane Central Register of Controlled Trials (Cochrane Library), EMBASE (Dialog), the Cumulative Index to Nursing and Allied Health Literature (CINAHL) (accessed via EBSCO), and APA PsycInfo (Ovid). In May 2021, we searched for ongoing and unpublished trials in trial registers such as ClinicalTrials.gov and the World Health Organization International Clinical Trials Platform Search Portal (WHO ICTRP), respectively. Details of these searches have been listed in the protocol (*Yoshihiro et al., 2021*). We conducted a 'snowball' search to identify studies that used reference lists of publications eligible for full-text review (including international guidelines) (*National Institute for Health and Care Excellence, 2009*; *Nolan et al., 2021*) and used Google Scholar to identify and screen those studies. We reconducted these searches in May 2022. Additionally, we contacted the authors of the original studies for unpublished or additional data.

### Population

We included trials with adult patients (age ≥18 years) admitted to ICUs and their informal caregivers; these trials were randomized during both ICU and hospital discharge. We included studies involving informal caregivers regardless of whether the admitted patient survived. We excluded studies involving patients and their caregivers who were younger than 18 years, did not provide consent for participation, or showed cognitive impairment. Furthermore, studies involving patients or caregivers who had experienced myocardial infarction or were in their perioperative period were excluded. In this article, we have referred to our target population of "critically-ill adult patients" as "patients." if not necessary.

### Interventions

We defined *intervention* as a service or program initiated after ICU discharge (within one month after hospital discharge), including multidisciplinary interventions, follow-up clinics, and other programs. In the included studies, we recognized counseling such as cognitive-behavioral therapy, that interventions target mental health conditions. In addition, we included psychological intervention performed as needed after monitoring. We incorporated all intervention periods by all professionals. In the included studies, nurses and physicians intervening in therapies had been trained for each study.

We excluded studies involving interventions in the ICU that were comprised of participant-led initiatives like ICU diaries and ICU records, interventions that provided general information pertaining to post-intensive care syndrome using web tools or video materials, or that compared enhanced physical rehabilitation with usual care. We did not predefine the details of the psychological interventions because we wanted to verify interventions that improved psychological outcomes other than physical and diary interventions.

### Outcomes

We included trials with defined clinical outcomes, such as symptoms of depression and PTSD, and all adverse events were considered primary outcomes among patients and caregivers (*Marra et al., 2018*). Additional outcomes among patients included anxiety, health-related quality of life (HR-QoL), pain, readmission, and long-term mortality; additional outcomes among caregivers included anxiety and HR-QoL. We followed core outcome sets (*Angus & Carlet, 2003*; *Major et al., 2016*; *Needham et al., 2017*). We selected outcomes for mental health as primary outcomes. We defined depression, PTSD, and anxiety as the prevalence rate of significant symptoms based on definitions by the included studies' authors, measured between three months and one year after randomization or ICU discharge. We defined adverse events using the incidence proportion of all adverse events set by the original authors during the follow-up period of included studies. We defined HR-QoL using a mental component summary of the Medical Health Survey Short-Form 36 (SF-36), measured between three months and one year after randomization or ICU discharge. SF-36 was used for self-reported evaluation scales for the evaluation of HR-QoL (*Angus & Carlet, 2003*; *Needham et al., 2017*). If the outcome of HR-QoL was measured by other self-reported evaluation scales in included studies, we assessed whether the scales

could be synthesized with SF-36. We defined pain using self-reported evaluation scales for pain set by the original authors, measured between three months and one year after randomization or ICU discharge. We defined readmission as the proportion of readmission (at least once) during the follow-up period of the included studies. For long-term mortality, we collected the reported mortality at the longest timepoint available in the study, which ranged between 3 and 12 months after randomization.

## Search strategy
### Selection process
Three reviewers (SY, YK, and KS) independently screened the titles and abstracts of records during the initial screening. We assessed records—included in the initial screening—for eligibility based on the inclusion criteria by reading the full texts. We resolved disagreements between two reviewers *via* discussion with a third reviewer (TS) to achieve consensus. We combined machine learning classifiers during the selection process (*Marshall et al., 2018*).

### Data collection process
Three reviewers (SY, YK, and KS) independently extracted data from the included studies using a standardized data collection form. We pre-checked the form by using 10 randomly selected studies. We extracted the following characteristics:

Methods: Study design, study follow-up period, and study country;

Participants: Country, setting, mental condition (depression, PTSD, and anxiety), sample size, age, relationship of informal caregivers with patients, and attrition;

Interventions: type, intervention about the psychological problem, providers, media, initiation, duration, and frequency;

Outcomes: primary and additional outcomes specified and collected, and the timepoints reported.

## Data items
### Study risk-of-bias assessment
Two to three reviewers (SY, YK, and KS) independently classified the risk of bias as "low", indicating "some concerns", or "high" based on the Risk-of-Bias 2.0 (*Sterne et al., 2019*). We resolved disagreements between two reviewers *via* discussion with the third reviewer (TS) to achieve consensus. As participants could not be blinded to the intervention owing to its nature, we assessed the overall risk-of-bias using four domains, which excluded the estimation of measurement-of-outcome.

### Effect measures
We analyzed the dichotomous variables by calculating risk ratios (RR) with 95% confidence intervals (CIs). We analyzed the continuous variables using standard mean differences (SMD) with 95% CI.

### Synthesis methods
We synthesized the collected variables (except for adverse events) using the random-effects method; data for patients and informal caregivers were synthesized separately. We used the Review Manager software (RevMan 5.4.2) for quantitative synthesis.

### Dealing with missing data

We used available data published and inquired to authors. We performed (modified) intention-to-treat data for all dichotomous data as much as possible. For continuous data, we did not impute missing data and performed a meta-analysis of the available data in the original studies and the converted data from available data based on the method in the Cochrane handbook (*Higgins et al., 2020*).

### Assessment of heterogeneity

We assessed heterogeneity by visual inspection of the forest plot and $I^2$ statistics ($I^2$ values of 0% to 40%: might not be important; 30% to 60%: may represent moderate heterogeneity; 50% to 90%: may represent substantial heterogeneity; 75% to 100%: considerable heterogeneity). We performed Cochrane $Chi^2$ test(Q-test) for $I^2$ statistic and defined *P* values less than 0.10 as statistically significant.

### Sensitivity analysis and subgroup analysis

We conducted the sensitivity analysis and subgroup analysis for the primary outcomes where sufficient data were available. We conducted sensitivity analysis of patients using studies measured by the Depression subscale of the Hospital Anxiety and Depression Scale(HADS-D) score for depression, studies measured by the Impact of Event Scale-Revised (IES-R) score for PTSD, and exclusion of imputed data. We conducted the sub-group analyses by timing for initiation of follow-up(in-hospital, out-hospital, or in- and out-hospital). For analysis for informal caregivers, we conducted sensitivity analysis using studies measured by IES-R scores for PTSD. We divided the ICU survivors and non-survivors in the sub-group analyses for informal caregivers.

### Reporting bias assessment

We identified the number of studies that had not been published on ClinicalTrials.gov and WHO ICTRP. We assessed outcome reporting bias by comparing the outcomes defined in trial protocols with the outcomes reported in the publications. We assessed the publication bias of outcomes by visual inspection of the funnel plots.

### Certainty assessment

Two reviewers (SY and TU) evaluated the certainty of evidence based on the Grading of Recommendations Assessment, Development and Evaluation (GRADE) approach (*Hultcrantz et al., 2017*). We resolved disagreements between two reviewers *via* discussion with the third reviewer (KY) to achieve consensus. We generated a table to summarize the findings of the seven outcomes (except for long-term mortality) using GRADE Pro GDT (https://gradepro.org) based on the Cochrane Handbook (*Higgins et al., 2020*). We selected the following outcomes for patients: (1) depression, (2) PTSD, (3) all adverse events, (4) anxiety, (5) HR-QoL, (6) pain, and (7) readmission. We selected the following outcomes for informal caregivers: (1) depression, (2) PTSD, (3) all adverse events, (4) anxiety, and (5) HR-QoL.

***Difference between protocol and review***

We did not conduct Egger's test as we synthesized data from fewer than 10 studies. We could not conduct planned sensitivity and sub-group analyses for PTSD and adverse events among patients and depression and adverse events among informal caregivers. We added a sub-group analysis for the endpoints of the measured outcomes, dividing them into 6 months and 12 months.

## RESULTS

### Study selection

We identified 10,425 records from databases and registers, and 46 records from citation searches and guidelines (*National Institute for Health and Care Excellence, 2009*; *Nolan et al., 2021*). After excluding duplicates, we could not retrieve the full text for one record from the Cochrane Library and confirmed that the record was an error through author inquiry. We assessed 240 full texts for eligibility and identified 119 studies. The flow diagram for study selection is presented in Fig. 1.

We identified six ongoing studies and one no-information study with patients, and one ongoing study with informal caregivers *via* ClinicalTrials.gov and WHO ICTRP. The details of all studies without results are outlined in Table S2. We excluded 92 studies after conducting full-text reviews; the reasons for their exclusion are listed in Table S3.

Since 12 of the included studies did not include results (*Chen et al., 2022*; *Ewens et al., 2019*; *Friedman et al., 2022*; *Gawlytta et al., 2020*; *Gawlytta et al., 2017*; *Haines et al., 2019*; *Khan et al., 2018*; *Moulaert et al., 2015*; *Ojeda et al., 2021*; *Rohr et al., 2021*) (NCT03431493, NCT03926533, NCT04329702), we included 15 studies for quantitative analysis. Of these 15 studies, 11 focused on patients (*Abdelhamid et al., 2021*; *Bloom et al., 2019*; *Cox et al., 2018*; *Cox et al., 2019*; *Cuthbertson et al., 2009*; *Daly et al., 2005*; *Douglas et al., 2005*; *Douglas et al., 2007*; *Hernández et al., 2014*; *Kredentser et al., 2018*; *McWilliams, Benington & Atkinson, 2016*; *Schmidt et al., 2016*; *Schmidt et al., 2020*; *Valsøet al., 2020*; *Vlake et al., 2021*), two focused on informal caregivers (*Ågren et al., 2019*; *Kentish-Barnes et al., 2017*), and two focused on both patients and informal caregivers (*Bohart et al., 2019*; *Jensen et al., 2016*; *Jones et al., 2004*; *Jones et al., 2003*). One study (*Cox et al., 2018*) was conducted with both patients and informal caregivers, but we could not retrieve outcome data for the informal caregivers. The details of these studies are outlined in Table 1.

### Study characteristics

We selected 13 studies that included 3,366 patients (Table 1A). These studies were conducted in eight countries: the USA ($n = 4$), the UK ($n = 3$), and Denmark, Germany, Norway, Netherlands, Canada, and Australia ($n = 1$ in each country). Patients in two studies had sepsis, and patients in six studies were provided mechanical ventilation. One study included patients with moderate PTSD symptoms after ICU discharge. Interventions in six studies focused on psychological problems among patients following critical illness. Interventions in seven studies included rehabilitation programs, multidisciplinary programs, and case management for monitoring and therapy for psychological problems.

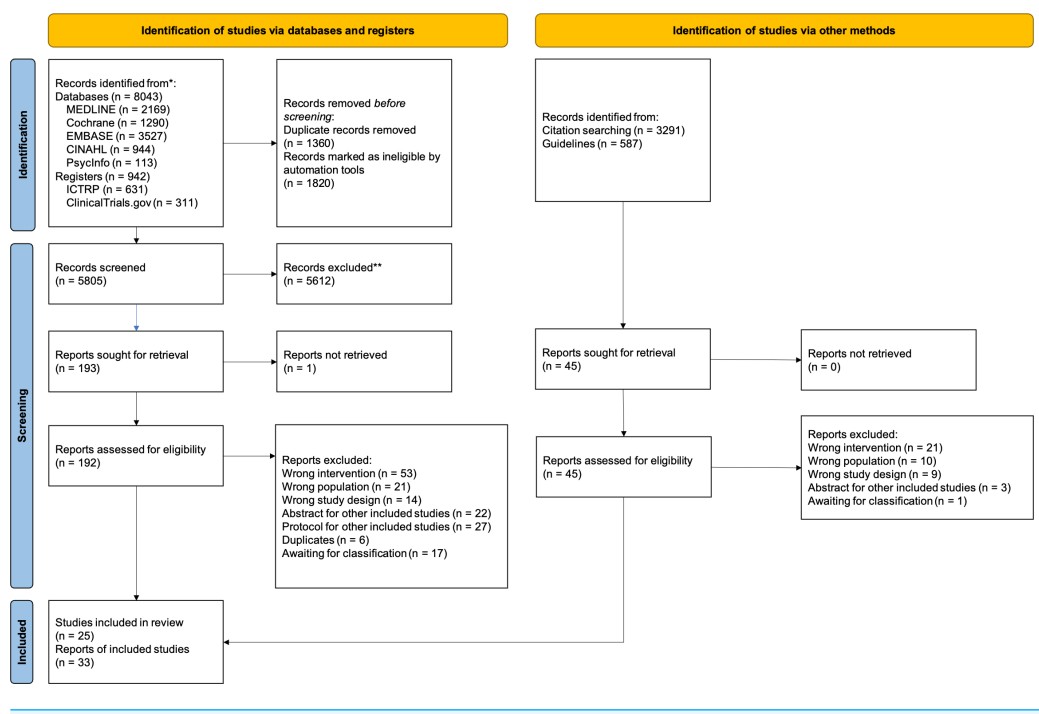

**Figure 1  PRISMA flow.**

We selected four studies, which included 538 informal caregivers (Table 1B). These studies were conducted in four countries: the UK, Denmark, France, and Sweden ($n = 1$ in each country). Most caregivers were spouses (47.8%), followed by children (16.8%), parents (9.3%), and siblings (1.3%) of the patients. All the studies included informal caregivers with or without psychological problems. Follow-ups were conducted on patients and caregivers in three studies, while one study conducted interventions on caregivers of the ICU non-survivor.

## Risk of bias in studies

The domains and overall risk of bias for each outcome are outlined in Fig S1. On the assessment of the randomization process, we found that one study (*Daly et al., 2005*) showed risk-of-bias concerns owing to no description of the details of concealment, and two studies (*Ågren et al., 2019*; *Bloom et al., 2019*) showed high risk of bias owing to an imbalance of patient characteristics. On the assessment of deviation from the intended interventions, we found that three studies (*Ågren et al., 2019*; *Cox et al., 2019*; *Daly et al., 2005*) showed some risk-of-bias concerns owing to the difference of drop-outs between each group, and one study (*Kredentser et al., 2018*) had a high risk of bias owing to no information and no conduct of modified intention for treatment. On the assessment of the missing outcome data, we found that four studies (*Cox et al., 2018*; *Cox et al., 2019*; *Jensen et al., 2016*; *Vlake et al., 2021*) had a low risk of bias for implementation of missing values; however, 10.2–52.1% of the participants dropped out in all eligible studies. The assessment of the outcome measurement indicated that all studies had a high risk of bias for outcomes estimated *via* self-reported questionnaires as patients could not be blinded to the interventions owing to their nature.

Peer J

**Table 1   Included studies.**

**(A) Patients**

| Authors year | Registry Number Country Observational period | No of participants Age, years Intervention/Control | Mental condition Intervention/Control | Attrition, % | Type of intervention | Type of intervention against psychological problem | Professionals/sources of intervention | Timing, duration, and/or frequency of intervention |
|---|---|---|---|---|---|---|---|---|
| *Jones et al. (2003)* | Not stated about registration the United Kingdom six months after ICU discharge | 69/57 Mean ± SD, 57 ± 17/59 ± 16 | Depression not stated; PTSD not stated; Anxiety not stated | 19 | Semi-structured programs for psychological, psychosocial, and physical problems | Provision of coping skills | Print media | After ICU discharge six weeks from one week |
| *Daly et al. (2005)* | No detail of registration the United States of America two months after hospital discharge | 231/103 Mean ± SD, 60.7 ± 16.6/ 61.4 ± 16.1 | Depression not stated; PTSD not stated; Anxiety not stated | 26 | Multidisciplinary intervention by nurse with support from a physician | Provision of coping skills | Nurse | After hospital discharge Two months |
| *Cuthbertson et al. (2009)* | ISRCTN24294750 The United Kingdom 12 months after ICU discharge | 143/143 Median (IQR), 59 (46–49)/60 (46–71) | Depression not stated; PTSD not stated; Anxiety not stated | 32.9 | Multidisciplinary intervention by nurse with support from an intensivist | Psychological intervention required after monitoring | Nurse | After hospital discharge Two times at 3 months and 9 months |
| *Jensen et al. (2016)* | NCT01721239 Denmark 12 months after ICU discharge | 190/196 Median (IQR), 66 (57.75–73.5)/67.5 (58–75) | Depression not stated; PTSD not stated; Anxiety not stated | 39.1 | Individualized, semi-structured program for psychological problem | Therapy: Cognitive behavioral therapy | Nurse | After ICU discharge Three times at 1–3, 5, and 10 months |
| *McWilliams, Benington & Atkinson (2016)* | NCT02491021 The United Kingdom seven weeks after hospital discharge | 37/36 Mean ± SD, 55.0 ± 12.9, 60.8 ± 12.3 | Depression not stated; PTSD not stated; Anxiety not stated | 13.7 | Rehabilitation program consisted of exercise and education component | Education | Nurse; Facilitators other than physician and nurse | After hospital discharge Total 6 educational sessions, 1 h per session, for 7 weeks |
| *Schmidt et al. (2016)* | ISRCTN61744782 Germany 12 months after ICU discharge | 148/143 Mean ± SD, 62.1 ± 14.1/ 61.2 ± 14.9 | Depression not stated; PTSD not stated; Anxiety not stated | 30.6 | Case management, telephone monitoring, and education of behavioral activation for patients, which consisted of general practitioner, case manager, and liaison physician | Provision of coping skills | Nurse; Physician | After ICU discharge Monthly for 6 months, and once every 3 months for the final 6 months |
| *Cox et al. (2018)* | NCT01983254 The United States of America 12 months after randomization (within two weeks after hospital discharge) | 39/47 Mean ± SD, 49.7 ± 13.8/53.7 ± 13.5 | *Patients* Depression 27/20 PTSD 4/6 Anxiety 24/17 | *Patients* 25.1 | Training for psychological problems, combined with Telephone and web | Provision of coping skills | Facilitators other than physician and nurse; Digital media | After hospital discharge six telephone sessions for thirty minutes, once per week |
| *Bloom et al. (2019)* | NCT03124342 The United States of America 30 days after hospital discharge | 145/157 Median (IQR), 56 (44–67), n = 111/56 (48–66), n = 121 | Depression Not stated; PTSD not stated; Anxiety Not stated | 27.5 | Multidisciplinary case management based on ICU recovery program | Psychological intervention required after monitoring | Nurse; Physician; Facilitators other than physician and nurse | After hospital discharge At least 30 days |
| *Cox et al. (2019)* | NCT02701361 The United States of America Three months after hospital discharge | 1) Telephone-based mindfulness training, 31/18 Mean ± SD, 48.1 ± 16.1/53.3 ± 12.6 | 1) Depression 4/1 PTSD 1/1 Anxiety 6/1 | 1) 10.2 | 1) Telephone-based training for psychological problems | 1) Provision of coping skills | 1) Facilitator other than physician and nurse | After hospital discharge Four sessions each week for one month |
|  |  | 2) Self-directed mindfulness training by mobile app, 31/18 Mean ± SD, 48.7 ± 15.3/53.3 ± 12.6 | 2) Depression 1/1 PTSD 2/0 Anxiety 2/1 | 2) 22.4 | 2) Self-directed training for psychological problems | 2) Provision of coping skills | 2) Digital media |  |
| *Kredentser et al. (2018)* | NCT02067559 Canada 90 days after ICU discharge | Sample size of usual care and psychoeducation in four arms 14/14 Mean ± SD, 59.3 ± 15.5/49.9 ± 16.9 | Depression not stated; PTSD not stated; Anxiety not stated | 60.7 | Education for psychological problem | Provision of coping skills | Print media | After ICU discharge or after return of the ability to provide consent |

Yoshihiro et al. (2023), *PeerJ*, DOI 10.7717/peerj.15260

**Table 1** (*continued*)

**(A) Patients**

| Authors year | Registry Number Country Observational period | No of participants Age, years Intervention/Control | Mental condition Intervention/Control | Attrition, % | Type of intervention | Type of intervention against psychological problem | Professionals/sources of intervention | Timing, duration, and/or frequency of intervention |
|---|---|---|---|---|---|---|---|---|
| *Valsøet al. (2020)* | NCT02077244 Kingdom of Norway Twelve months after ICU discharge | 111/113 Mean ± SD, 53 ± 16/50 ± 18 | Depression not stated; PTSD 111/113; Anxiety not stated | 23.7 | Individualized, semi-structured program for psychological and psychosocial problems | Therapy: Cognitive behavioral therapy | Nurse | After ICU discharge three times in the first week, one and two months later |
| *Abdelhamid et al. (2021)* | ACTRN12616000206426 Australia six months after hospital discharge | 21/21 Mean ± SD, 64 ± 11/68 ± 8 | Depression not stated; PTSD not stated; Anxiety not stated | 38.1 | Multidisciplinary intervention by an intensivist and endocrinologist | Psychological intervention required after monitoring | Physician | After hospital discharge At least one time, repeated as needed for six months from one month |
| *Vlake et al. (2021)* | NL6611 Netherlands six months after ICU discharge | 25/25 Median (95% range), 61 (23–75)/59 (59–80) | Depression 6/12 PTSD 12/13; Anxiety not stated | 16 | ICU-specific virtual reality for psychological problem | Therapy: Virtual reality exposure therapy | Digital media | After ICU discharge The number of desired sessions was offered daily |

**(B) Informal caregivers**

| Authors year | Registry number Country Observational period | No of participants Age, years Intervention/Control | Mental condition Intervention/Control | Relationship of caregivers with patients | | | | Attrition, % | Type of intervention | Type of intervention against psychological problem | Professionals/sources of intervention; | Timing, duration, and/or frequency of intervention |
|---|---|---|---|---|---|---|---|---|---|---|---|---|
| | | | | Spouse, % | Child, % | Parent, % | Sibling, % | | | | | |
| *Jones et al. (2004)* | No detail of registration the United Kingdom six months after ICU discharge | Caregivers 58/46 Mean ± SD, 62 ± 17/60 ± 15.4 | Depression 13/14 PTSD not stated; Anxiety 34/29 | 51.9 | 19.2 | 18.3 | 6.7 | 19.2 | Training for psychological problems | Provision of coping skills | Print media; | After ICU discharge six weeks from one week |
| *Jensen et al. (2016)* | NCT03264365 Denmark 12 months after ICU discharge | 87/94 Median (IQR), 57.4 (50–67)/61 (41.8–69) | Depression not stated; PTSD not stated; Anxiety not stated | 71.3 | Not stated | 17.1 | Not stated | 38.7 | Individualized, semi-structured program for psychological problems | Therapy: Cognitive behavioral therapy | Nurse; | After ICU discharge Once at 1–3 months |
| *Kentish-Barnes et al. (2017)* | NCT02325297 France six months after in the 24 h following the death of the patient | 109/99 Median (Range), 57 (46–65.5) /56 (44–64.5) | Depression not stated; PTSD not stated; Anxiety not stated | 35.6 | 39.9 | Not stated | Not stated | 22.3 | Condolence letters | Empathy: Condolence letters | Print media; | After patient's death Once at 15 days |
| *Ågren et al. (2019)* | NCT03325049 The Kingdom of Sweden 12 months after ICU discharge | Seven families (17 individuals) /10 families (28 individuals) Mean ± SD, 60 ± 19/61 ± 17 | Depression not stated; PTSD not stated; Anxiety not stated | Not stated | Not stated | Not stated | Not stated | 51.1 | Health-promoting conversation forced on experience of the current situation | Empathy: Counseling | Nurse; | After ICU discharge Two weeks interval, within approximately 4 to 8 weeks after hospital discharge |

**Notes.**

IQR, Interquartile range; ICU, intensive care unit; SD, standard deviation; PTSD, post-traumatic stress disorder.

## Patient outcomes
### Depression
As shown in Fig. 2 and Table 2, ICU follow-ups resulted in little to no differences in the prevalence rate of depressive symptoms among patients (RR 0.89, 95% CI [0.59, 1.34]; $I^2 = 1\%$; four studies, 758 patients; low-certainty evidence) (*Abdelhamid et al., 2021*; *Jensen et al., 2016*; *Jones et al., 2003*; *Schmidt et al., 2016*; *Vlake et al., 2021*); we detected slight heterogeneity. Planned sensitivity analyses of studies using the Depression subscale scores of the Hospital Anxiety and Depression Scale (HADS-D) yielded similar findings (RR 0.90, 95% CI [0.50–1.63]). Planned sensitivity analysis that excluded the imputed data showed a similar trend (RR 1.08, 95% CI [0.55–2.09]). Sub-group analysis for the timing of follow-up initiation showed a similar trend in the group of initiation from both ICU discharge and hospital discharge. In the sub-group analysis, there was no difference in the endpoint to measure depressive symptoms between 6 months and 12 months. Details of the analysis are provided in Fig S2.

### Post-traumatic stress disorder
ICU follow-ups resulted in little to no differences in the prevalence rate of PTSD symptoms among patients (RR 0.84, 95% CI [0.55–1.30]; $I^2 = 53\%$; four studies, 732 patients; low-certainty evidence) (*Jensen et al., 2016*; *Jones et al., 2003*; *Schmidt et al., 2016*; *Vlake et al., 2021*); we detected moderate heterogeneity (Fig. 2 and Table 2). Planned sensitivity analysis of studies using the Impact of Event Scale- Revised scores (IES-R) yielded similar results (RR 0.51, 95% CI [0.08–3.23]). The planned sensitivity analysis that excluded the imputed data generated similar findings (RR 1.06, 95% CI [0.75–1.50]; Fig. S2). Sub-group analysis for the endpoint to measure PTSD symptoms showed a similar trend in the endpoint to measure depressive symptoms between 6 months and 12 months. Details of the analysis are provided in Fig. S2.

### Adverse events
Although evidence indicates considerable uncertainty, ICU follow-ups resulted in little to no differences in the occurrence of adverse events (*Vlake et al., 2021*) (Table 2). Two studies included adverse events as outcome measures (*Bloom et al., 2019*; *Vlake et al., 2021*). One published article (*Bloom et al., 2019*) did not report the results pertaining to adverse events, and we could not obtain information about adverse events from its authors. This study defined adverse events as the need for intervention to prevent events such as mortality, prolonged hospitalization, acquisition of disability, congenital anomalies, and birth defects. Another study (*Vlake et al., 2021*) defined adverse events as incidents of cybersickness, delirium, or the use of haloperidol. Considering the clinical heterogeneity in studies, we included all types of adverse events except for cybersickness.

### Anxiety
ICU follow-ups resulted in little to no differences in the prevalence rate of anxiety symptoms among patients (RR 1.04, 95% CI [0.68–1.60]; $I^2 = 0\%$; two studies, 488 patients; low certainty of evidence) (*Jensen et al., 2016*; *Jones et al., 2003*); no significant heterogeneity was detected (Table 2 and Fig. S3).

A)

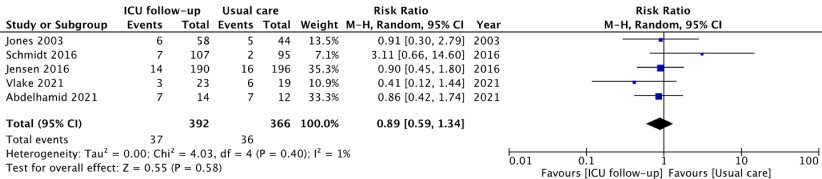

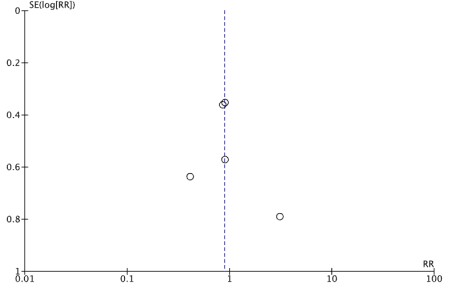

B)

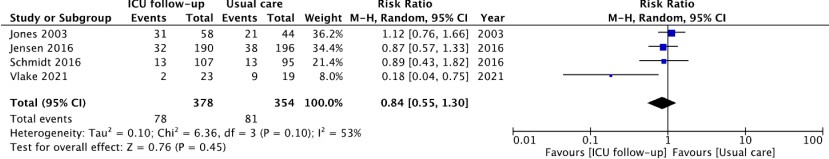

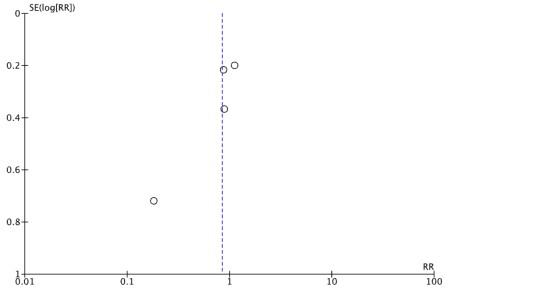

**Figure 2  Forest plot and funnel plot of primary outcomes for patients.** (A) Depression, (B) Post-traumatic stress disorder. Adverse events were not pooled.

### Health-related quality of life

ICU follow-ups resulted in little to no differences in the HR-QoL scores among patients (SMD 0.05, 95% CI [−0.08–0.18]; $I^2 = 0$%; seven studies, 905 patients; low-certainty evidence) (*Abdelhamid et al., 2021*; *Cox et al., 2018*; *Cox et al., 2019*; *Cuthbertson et al., 2009*; *Jensen et al., 2016*; *Schmidt et al., 2016*; *Vlake et al., 2021*); no significant heterogeneity was detected (Table 2 and Fig. S3). Of the seven studies, four measured the HR-QoL using the Mental Component Summary (MCS) of the Short-Form-36 (SF-36) (*Abdelhamid et al., 2021*; *Cuthbertson et al., 2009*; *Jensen et al., 2016*; *Schmidt et al., 2016*), one study used the MCS of the SF-12 (*Vlake et al., 2021*), and two studies used the EuroQoL Visual Analogue

**Table 2  Summary of findings for patients.**

**ICU follow-up compared to usual care for critically ill patients**

**Patient or population:** Critically ill patients

**Setting:**

**Intervention:** ICU follow-up

**Comparison:** Usual care

| Outcomes | Anticipated absolute effects[*] (95% CI) | | Relative | No of | Certainty of | Comments |
|---|---|---|---|---|---|---|
| | **Risk with Usual care** | **Risk with ICU follow-up** | (95% CI) | (studies) | (GRADE) | |
| Proportion of patients with depression | Median 114 per 1,000 | 101 per 1,000 (67 to 152) | RR 0.89 (0.59 to 1.34) | 758 (5 RCTs) | ⊕⊕◯◯ Low[a,b] | |
| Proportion of patients with PTSD | Median 145 per 1,000 | 122 per 1,000 (80 to 188) | RR 0.84 (0.55 to 1.30) | 732 (4 RCTs) | ⊕⊕◯◯ Low[a,b] | |
| All adverse events | Median 0 per 1,000 | 0 per 1,000 (0 to 0) | Not estimable | 42 (1 RCT) | ⊕◯◯◯ Very low[a,c] | |
| Proportion of patients with anxiety | Median 206 per 1,000 | 214 per 1,000 (140 to 329) | RR 1.04 (0.68 to 1.60) | 488 (2 RCTs) | ⊕⊕◯◯ Low[a,b] | |
| HR-QoL | – | SMD 0.05 higher (0.08 lower to 0.18 higher) | – | 905 (8 RCTs) | ⊕⊕◯◯ Low[a,b] | |
| Pain | – | SMD 0.08 lower (0.32 lower to 0.17 higher) | – | 258 (3 RCTs) | ⊕⊕◯◯ Low[a,b] | |
| Readmission | Median 274 per 1,000 | 261 per 1,000 (211 to 318) | RR 0.95 (0.77 to 1.16) | 1016 (8 RCTs) | ⊕⊕◯◯ Low[a,b] | |

[*]**The risk in the intervention group** (and its 95% confidence interval) is based on the assumed risk in the comparison group and the **relative effect** of the intervention (and its 95% CI).

Confidence interval, CI; health-related quality of life; HR-QoL; intensive care unit, ICU; odds ratio; OR; risk ratio RR; standardized mean difference, SMD; post-traumatic stress disorder, PTSD; randomized controlled trial, RCT.

**GRADE Working Group grades of evidence**

**High certainty:** we are very confident that the true effect lies close to that of the estimate of the effect.

**Moderate certainty:** we are moderately confident in the effect estimate: the true effect is likely to be close to the estimate of the effect, but there is a possibility that it is substantially different.

**Low certainty:** our confidence in the effect estimate is limited: the true effect may be substantially different from the estimate of the effect.

**Very low certainty:** we have very little confidence in the effect estimate: the true effect is likely to be substantially different from the estimate of effect.

Notes.
[a]Downgrade for a high risk of bias: Some included studies assessed presented some concerns.
[b]Downgrade for imprecision: The sample size was small.
[c]Downgrade for imprecision: Outcome was reported in only 1 study.

Scale (EQ-VAS) (*Cox et al., 2018*; *Cox et al., 2019*). The analysis of studies using the MCS of the SF-36 and the SF-12 yielded similar findings (SMD 0.04, 95% CI [−0.11–0.19]).

### Pain

ICU follow-ups resulted in little to no differences in the pain scores among patients (SMD −0.08, 95% CI [−0.32, 0.17]; $I^2 = 0\%$; three studies, 258 patients; low-certainty evidence) (*Abdelhamid et al., 2021*; *Schmidt et al., 2016*; *Vlake et al., 2021*); no significant heterogeneity was detected (Table 2 and Fig. S3). One study (*Schmidt et al., 2016*) measured pain intensity using the Graded Chronic Pain Scale; one study (*Abdelhamid et al., 2021*)

used the pain comportment of the SF-36. For one study (*Vlake et al., 2021*), we obtained data for the pain comportment of the SF-12 which was converted to the VAS 100 scale *via* author inquiry.

### Readmission

ICU follow-ups resulted in little to no significant in the proportion of patients readmitted to the hospital during follow-up periods (RR 0.95, 95% CI [0.77–1.16]; $I^2 = 18\%$; seven studies, 1,016 patients; low certainty evidence) (*Abdelhamid et al., 2021*; *Bloom et al., 2019*; *Cox et al., 2018*; *Cox et al., 2019*; *Daly et al., 2005*; *Jensen et al., 2016*; *McWilliams, Benington & Atkinson, 2016*); no significant heterogeneity was detected (Table 2 and Fig. S3).

### Long term mortality

ICU follow-ups resulted in little to no differences in long-term mortality among patients (RR 0.95, 95% CI [0.74–1.21]; $I^2 = 0\%$; nine studies, 1,608 patients) (*Abdelhamid et al., 2021*; *Cox et al., 2018*; *Cuthbertson et al., 2009*; *Jensen et al., 2016*; *Jones et al., 2003*; *Kredentser et al., 2018*; *Schmidt et al., 2016*; *Valsøet al., 2020*; *Vlake et al., 2021*) (Fig. S3); no significant heterogeneity was detected.

## Informal caregiver outcomes
### Depression

Although the evidence indicated considerable uncertainty, ICU follow-ups increased the prevalence rate of depressive symptoms—measured using the HADS-D—among informal caregivers (RR 1.58 95% CI [1.01–2.46]; one study, 188 caregivers; very low-certainty evidence) (*Kentish-Barnes et al., 2017*) (Table 3). However, the other two studies (*Bohart et al., 2019*; *Cox et al., 2018*) did not report the proportion of informal caregivers with depressive symptoms, but instead provided their HADS-D scores. The point estimate of HADS-D score was higher in the ICU follow-up groups than control; thus, no inconsistencies were observed.

### Post-traumatic stress disorder

Although the evidence indicated considerable uncertainty, ICU follow-ups increased the prevalence rate of PTSD symptoms—measured using the IES-R—among informal caregivers (RR 1.36, 95% CI [0.91–2.03]; $I^2 = 19\%$; two studies, 303 caregivers; very low certainty of evidence) (*Bohart et al., 2019*; *Kentish-Barnes et al., 2017*) (Fig. 3 and Table 3); we detected slight heterogeneity. Planned sensitivity analysis of studies using the IES-R showed that ICU follow-ups significantly increased the proportion of patients with PTSD(RR 1.51, 95% CI [1.09–2.09]) (Fig. S4). One study (*Cox et al., 2018*) measured the IES-R scores and not the proportion of informal caregivers with PTSD; the point estimate of the IES-R scores was higher for the ICU follow-up group. In a sub-analysis, we found that only caregivers with non-survivors developed PTSD owing to ICU follow-ups (Fig. S4). In another sub-analysis, there was no difference in the endpoint to measure PTSD symptoms between 6 and 12 months.

### Adverse events

Eligible studies with informal caregivers did not define any adverse events (Table 3).

**Table 3  Summary of findings for informal caregivers.**

**ICU follow-up compared to usual care for caregivers of critically ill patients**

**Patient or population:** Caregivers of critically ill patients

**Setting:**

**Intervention:** ICU follow-up

**Comparison:** Usual care

| Outcomes | Anticipated absolute effects[*] (95% CI) | | Relative | No of | Certainty of | Comments |
|---|---|---|---|---|---|---|
| | **Risk with Usual care** | **Risk with ICU follow-up** | (95% CI) | (Studies) | (GRADE) | |
| Proportion of caregivers with depression | Median 242 per 1,000 | 382 per 1,000 (244 to 595) | RR 1.58 (1.01 to 2.46) | 188 (1 RCT) | ⊕◯◯◯ Very low[a,b] | |
| Proportion of caregivers with PTSD | Median 352 per 1,000 | 478 per 1,000 (320 to 714) | RR 1.36 (0.91 to 2.03) | 303 (2 RCTs) | ⊕◯◯◯ Very low[a,b] | |
| All adverse events | Not pooled | Not pooled | Not pooled | (0 RCTs) | – | |
| Proportion of caregivers with anxiety | Median 318 per 1,000 | 372 per 1,000 (264 to 518) | RR 1.17 (0.83 to 1.63) | 272 (2 RCTs) | ⊕◯◯◯ Very low[a,b] | |
| HR-QoL | – | SMD 0.07 lower (0.41 lower to 0.27 higher) | - | 133 (2 RCTs) | ⊕◯◯◯ Very low[a,c] | |

[*]**The risk in the intervention group** (and its 95% confidence interval) is based on the assumed risk in the comparison group and the **relative effect** of the intervention (and its 95% CI).

Confidence interval, CI; health-related quality of life; HR-QoL; intensive care unit, ICU; risk ratio RR; standardized mean difference, SMD; post-traumatic stress disorder, PTSD; randomized controlled trial, RCT.

**GRADE Working Group grades of evidence**

**High certainty:** we are very confident that the true effect lies close to that of the estimate of the effect.

**Moderate certainty:** we are moderately confident in the effect estimate: the true effect is likely to be close to the estimate of the effect, but there is a possibility that it is substantially different.

**Low certainty:** our confidence in the effect estimate is limited: the true effect may be substantially different from the estimate of the effect.

**Very low certainty:** we have very little confidence in the effect estimate: the true effect is likely to be substantially different from the estimate of effect.

Notes.
[a]Downgrade for a high risk of bias: This intervention was not able to blind the assessors because of both the nature of intervention and the use of self-reported outcomes.
[b]Downgrade for imprecision: The sample size was small.
[c]Downgrade for imprecision: CI included possibility of both reasonable benefit and harm.

### Anxiety

Although the evidence indicated considerable uncertainty, ICU follow-ups increased the prevalence rate of anxiety symptoms, measured using the Anxiety subscale of the HADS (HADS-A), among informal caregivers (RR 1.17, 95% CI 0.83 to 1.63; two studies, 272 caregivers; very low-certainty evidence) (*Jones et al., 2004*; *Kentish-Barnes et al., 2017*) (Table 3 and Fig. S5); no significant heterogeneity was detected ($I^2 = 0\%$). One study (*Cox et al., 2018*) measured the HADS-A scores and not the proportion of caregivers with anxiety; the point estimate of the HADS-A scores was higher for the ICU follow-up group.

### Health-related quality of life

Although the evidence indicated considerable uncertainty, ICU follow-ups had little to no effect on the HR-QoL measured using the MCS of the SF-36 among informal caregivers (MD −0.70, 95% CI [−4.51, 3.11]; $I^2 = 0\%$; two studies, 133 caregivers; very low certainty

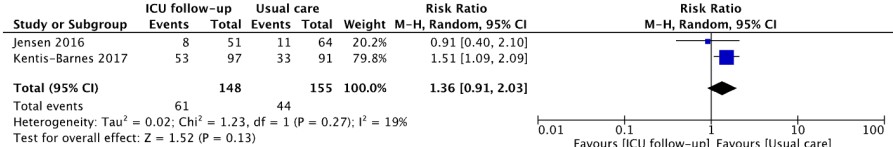

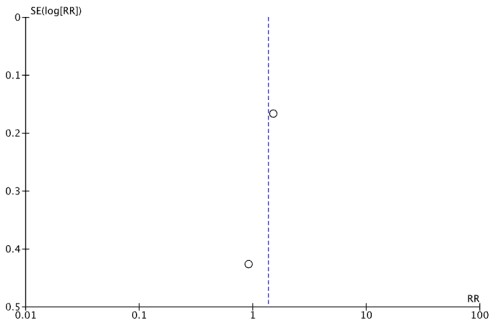

**Figure 3 Forest plot and funnel plot of primary outcomes for informal caregivers.** Post-traumatic stress disorder. Since the outcome of depression was reported in only 1 RCT, we do not show the forest plot and funnel plot. Adverse events were not pooled.

of evidence); no significant heterogeneity was detected (*Ågren et al., 2019*; *Bohart et al., 2019*) (Table 3 and Fig. S5).

## DISCUSSION

Our SR/MA revealed that ICU follow-ups did not decrease the prevalence of depression, PTSD, and anxiety among patients. On the contrary, ICU follow-ups increased the prevalence of depression and PTSD among informal caregivers; however, there was low certainty of evidence. Furthermore, sensitivity and sub- analyses yielded similar results. Although the certainty of the evidence was low, the ICU follow-up did not decrease pain among patients.

The follow-up initiated after ICU discharge did not reduce psychological dysfunction among critically-ill patients. A Cochrane SR focusing on ICU survivors included four RCTs and concluded that the evidence for the efficacy of post-ICU follow-ups was insufficient (*Schofield-Robinson et al., 2018*). Our SR/MA revealed the ineffectiveness of post-ICU follow-ups for depression and anxiety with greater certainty than the Cochrane SR (*Schofield-Robinson et al., 2018*). The National Institute for Health and Clinical Excellence guidelines (*National Institute for Health and Care Excellence, 2009*) suggested that medical staff should conduct psychological intervention to monitor and develop preventive or treatment strategies for psychological dysfunction. However, our findings contradicted this guideline. Two reasons may explain this finding. First, the intervention content differed. The guideline (*National Institute for Health and Care Excellence, 2009*) was based on interventions comprised of enhanced or individualized physical rehabilitation; however, we focused on psychological intervention and excluded interventions pertaining to

mobilization. Second, the timings of initiation of interventions were different. The guideline (*National Institute for Health and Care Excellence, 2009*) suggested that medical staff might be suitable to assess the need for patient rehabilitation before ICU discharge; however, we focused on interventions initiated after ICU discharge and interventions for psychological dysfunction. Considering our findings, follow-ups focusing on psychological intervention initiated after ICU discharge need not be conducted for patients.

The current approaches to psychological intervention after ICU discharge were not helpful for patients and led to increased depression, PTSD, and anxiety in informal caregivers. Patients and informal caregivers have high levels of depression, anxiety, and PTSD, and the current approaches fail to address this, though it is important to screen for all components of PICS. The guidelines published by the European Resuscitation Council and the European Society of Intensive Care Medicine pertained to cardiac arrests among adults (*Nolan et al., 2021*). Based on qualitative synthesis, the guideline panel suggested that medical staff should monitor and provide information about psychological problems among informal caregivers following patients' hospital discharge (*Nolan et al., 2021*). Our SR scoped the only RCTs as a more rigorous study design with narrower eligible criteria than that of the previous SR (*Rosa et al., 2019*). As for the effect of ICU follow-up on psychological symptoms, our meta-analysis conclusions contradicted that of the previous SRs accordingly (*Cherak et al., 2021*; *Rosa et al., 2019*). This could be because of the differences in the target informal caregivers as well as the different design used in the two SRs. A recent SR showed that mental health interventions after ICU discharge may alleviate psychological problems among informal caregivers (*Cherak et al., 2021*). The primary relationship between informal caregivers and patients in the previous SR was that of parents of children. The primary informal caregivers of critically ill adults in our SR/MA were spouses, so the intervention to reduce psychological modulation in our SR was different from that of the previous SR. Moreover, the SR included quasi-experimental and uncontrolled trials and did not conduct sub-analyses of the relationship with patients. These reasons could lead to negative results. Although it is necessary to monitor psychological dysfunction among informal caregivers, follow-ups might have both positive and harmful effects on depression, PTSD, and anxiety among informal caregivers (after the ICU discharge) of adult patients.

Further research must generate a risk assessment model and other interventions to reduce psychological dysfunction and alleviate the intensity of risk factors among patients and their informal caregivers in the high-risk group. The prevalence of depression and PTSD among patients in the usual care group in our SR/MA was lower after 12 months from ICU discharge compared to patients in previous reviews (*Parker et al., 2015*; *Rabiee et al., 2016*). Furthermore, although the guidelines (*National Institute for Health and Care Excellence, 2009*) suggested the need for risk assessment of psychological dysfunction among critically-ill patients, we find no risk assessment model suitable for psychological dysfunction. Previous studies showed that pain was associated with psychological dysfunction among patients in the ICU (*Puntillo et al., 2018*) and persisted after ICU discharge (*Kemp et al., 2019*); thus, pain could be one of the risk factors for psychological dysfunction. It is unclear whether follow-up would reduce pain or the risk (of psychological dysfunction)

associated with factors like pain. Additionally, our eligible studies excluded patients with cognitive impairments due to the nature of the intervention. One cohort study reported that symptoms of PICS overlapped (*Marra et al., 2018*). Patients and their informal caregivers with cognitive impairments might not be able to find and avoid psychological intervention by themselves. Thus, in a future study, we should develop an effective intervention for participants with a high-risk of PICS.

Our SR/MA had several strengths. First, we searched databases like APA PsycInfo (Ovid), which covered the psychiatric domain, in addition to guidelines and citations *via* Google Scholar. Second, we conducted sensitivity and sub-analysis based on pre-registered protocols, yielding interesting findings. However, we could not verify the results for all primary outcomes owing to the small number of eligible studies. Third, several studies included in this SR/MA were well-designed except for the nature of the intervention. Finally, our definitions for the critical outcome measures were based on core outcomes among critically ill patients.

However, several limitations of our SR/MA need to be acknowledged. First, our search strategy involved using keywords for outcome measures instead of intervention strategies. Searches using outcome keywords might result in more favorable outcomes for intervention (*Tsujimoto et al., 2021*). Nevertheless, our SR/MA found negative results for the effectiveness of ICU follow-ups. Second, the attrition of participants in all eligible studies was higher than 20%. As participants who developed psychological dysfunction tended to withdraw from the studies, the compliance of participants with the needs of follow-ups decreased. Finally, there were several issues that require further investigation. Most reviewed studies did not report adverse events, which was a critical outcome measure for ICU survivors and their families. We could not verify the effective initiation, period, and type of intervention as they were outside the scope of our SR/MA. Similarly, the researchers' experiences were unknown.

## CONCLUSION

We conducted a systematic review and meta-analysis for ICU follow-ups initiated after ICU discharge, focusing on psychological intervention. We found that ICU follow-ups did not decrease the risk of psychological dysfunction and readmission among patients. The evidence of the effect of ICU follow-up on adverse events among patients was insufficient. Similarly, there was insufficient evidence for the effect of ICU follow-ups among informal caregivers. Future studies should focus on ICU follow-ups for high-risk patients and informal caregivers of surviving patients to monitor in order to prevent the development of psychological dysfunction.

## ACKNOWLEDGEMENTS

We greatly appreciate Sara L. Douglas, Ronald L. Hickman Jr., Johan H. Vlake, V.R.M. Moulaert, Kimberley Haines, Konrad Schmidt, Nancy Kentish-Barnes, Christina Jones, and Janet F Jensen for providing additional information about their studies. We also appreciate

Koichi Mino for providing information for this study. The authors greatly appreciate Editage for English language editing.

### Funding

This work was supported by JSPS KAKENHI Grant Number JP18K17719. The funders had no role in study design, data collection and analysis, decision to publish, or preparation of the manuscript.

### Grant Disclosures

The following grant information was disclosed by the authors:
JSPS KAKENHI: JP18K17719.

### Competing Interests

Shunsuke Taito, Yusuke Tsutsumi, and Yuki Kataoka are affiliated Scientific Research WorkS Peer Support Group (SRWS-PSG), Osaka, JAPAN, which is an academic research group. Kota Yamauchi is employed by the Steel Memorial Yawata Hospital. Yuki Kataoka is employed by the Kyoto Min-iren Asukai Hospital. Yusuke Tsutsumi is employed by the National Hospital Organization Mito Medical Center. The authors declare there are no competing interests.

### Author Contributions

- Shodai Yoshihiro conceived and designed the experiments, performed the experiments, analyzed the data, prepared figures and/or tables, and approved the final draft.
- Shunsuke Taito conceived and designed the experiments, performed the experiments, authored or reviewed drafts of the article, and approved the final draft.
- Kota Yamauchi conceived and designed the experiments, performed the experiments, analyzed the data, prepared figures and/or tables, and approved the final draft.
- Shunsuke Kina conceived and designed the experiments, performed the experiments, analyzed the data, prepared figures and/or tables, and approved the final draft.
- Takero Terayama conceived and designed the experiments, prepared figures and/or tables, and approved the final draft.
- Yusuke Tsutsumi conceived and designed the experiments, authored or reviewed drafts of the article, and approved the final draft.
- Yuki Kataoka conceived and designed the experiments, authored or reviewed drafts of the article, and approved the final draft.
- Takeshi Unoki conceived and designed the experiments, authored or reviewed drafts of the article, and approved the final draft.

### Data Availability

   The raw measurements are available in the Supplemental Files.

## Supplemental Information

Supplemental information for this article can be found online at http://dx.doi.org/10.7717/peerj.15260#supplemental-information.

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
