# Peer review of "Follow-up focused on psychological intervention initiated after intensive care unit in adult patients and informal caregivers: a systematic review and meta-analysis"

_PeerJ, doi:10.7717/peerj.15260_

## Round 0.1 · original submission · Major Revisions

This is a potentially relevant and interesting review, which was also acknowledged by the reviewers. Before the review can be published a major revision is required. In particular, I would like to ask authors to pay attention to the following points:

1) It is not clear what is included in 'psychological intervention', when it starts and when it ends.

2) the recommendations are not fully supported by the data, please carefully reconsider your suggestions

3) The language needs some attention. PeerJ does not provide language editing, please have the manuscript reviewed by a professional editor or proficient English speaker.

4) Please provide a point-by-point response to the feedback.

Reviewer 1 ·

Basic reporting

The introduction gives an overview of the consequences of a stay in the intensive care unit, The background gives an overview of the complex issues. However, there is a lack of infor-mation and literaure on PICS F. It is also unclear what is meant by psychological interventions - and do these interventions relate to the ICU stay (such as diary) or after discharge? Are there differences in interventions during follow-up services between patients and infor-mal caregiv-ers? Follow-ups are described differently in the literature in terms of location, interval or inter-ven-tion. What is a follow-up services in the review? The article is structured in a comprehensi-ble and understandable way. The tables and figures are numbered and clearly arranged.

Experimental design

The research question is well defined.
The published protocol of the study illustrates the method. - and give information to replicate.
However, were all studies on follow-up included or only those with a focus on psychological interventions? Were studies included only in which the interventions took place during the ICU stay or only afterwards

Validity of the findings

The study shows what knowledge has been newly generated .
Questions about study selection are open (Kentish –Barnes et al. 2017 included in their study family members following the death of the patients. Is this study comparable to the other stud-ies?)
The authors have shown in their analysis that ICU follow-Up did not decrease the prevalence of de-pression or PTSD. You explain dies contrary results- But to what extent are these symp-toms (Depression, PTSD) part of the normal course from a Critical Illness and ICU stay? Which courses do they show and when do ICU follow-up interventions make sense at all? - and what are the individual patterns / variation of the intervention groups within the patients (de-pression, PTSD, HRQol)

Additional comments

Thank you for providing me the opportunity to review your manuscript. This is an important topic for the multiprofessional team. I have some concerns to improve the manuscript
Your introduction gives an overview of the consequences of a stay in the intensive care unit, but your introduction need more information’s.
On the part of the patients, the PICS is presented by psychological dysfunction, cognitive im-pairment and muscle weakness. Is muscle weakness the only physical limitation'? - On the part of informal caregivers, there is a lack of information on PICS F? What exactly do you mean by psychological interventions? – and do these interventions relate to the ICU stay (such as diary) or after discharge? Are there differences in interventions during follow-up services between patients and informal caregivers? What the evidence says? Please explain in the introduction section, whether you focus on adult patients or children in your review?
Eligibility criteria:
Were all studies on follow-up included or only those with a focus on psychological interven-tions?- Were studies included only in which the interventions took place during the ICU stay or only afterwards?
Follow-ups are described differently in the literature in terms of location, interval or interven-tion. What is a follow-up services in your review?
Please explain in the methods section why you specially excluded patients with myocardial in-farction, these patients can make complications and suffer from a PICS.
Intervention:
I thank you for explaning the term Follow-Up. Please tell readers which profession had to carry out the intervention to be considered as a psychological intervention.
Outcomes:
You selected outcomes for mental health as primary outcomes, such as depression or PTSD, these studies investigate PTSD or symptoms of PTSD?
Data Collection
What experience do the researchers have?
Outcomes: The selected outcomes for mental health are depression or PTSD, - these studies investigate PTSD or symptoms of PTSD?
Intervention: Which profession had to carry out the intervention to be considered as a psycho-logical intervention.
Study selection
Kentish –Barnes et al. 2017 included in their study family members following the death of the patients. Is this study comparable to the other studies?
Patients Outcomes / Results:
Posttraumatic stress disorder with patients and families members. See comments above.
Diskussion:
You have shown in your analysis that ICU follow-Up did not decrease the prevalence of de-pression or PTSD. You explain dies contrary results- But to what extent are these symptoms (Depression, PTSD) part of the normal course from a Critical Illness and ICU stay? Which courses do they show and when do ICU follow-up interventions make sense at all? - and what are the individual patterns / variation of the intervention groups within the patients (depression, PTSD, HRQol)
Line 437 - …Further research must generate a risk assessment model and other interventions to reduce psychological dysfunction. What kind of interventions have been thought of here?
Your research questions was: .Does follow-up with patients and their informal caregivers after ICU discharge improve mental health compared to usual care?. What does the evidence says to the timepoint?
Conclusion:
Line 474: Similarly, they were not effective in alleviating pain and or improving patients. HRQoL. Were interventions investigated to reduce pain or improve HRQoL?
Table 1: what does “Staff delivering intervention” mean?
The study shows what knowledge has been newly generated .

Reviewer 2 ·

Basic reporting

There are a few grammatical errors throughout the manuscript.

Framing of PICS is somewhat limited and as presented. Mental health is just one component of PICS, as written in a few sentences in the introduction, suggests it is the entirety of PICS

Experimental design

Would consider adding adult patients to clarify the adult exclusive population based on discussion in the introduction regarding the Cherak SR.

Three months to 12 months is significant endpoint heterogeneity, but is not reflected in the heterogeneity testing because this section seems to indicate that all endpoints were treated equally whether they were measured at 3 months, 6 months, or 12 months. We know that, for example PTSD rates with regards to time vary. Point Prevalence estimates stratified by time would help to strengthen this analysis and help determine if the negative results of this study are due to the dilution of endpoints.

Population heterogeneity not addressed fully. In the study characteristics the authors report ICU diagnose for 8/13 included studies. An expanded discussion regarding ICU diagnoses is needed as a consideration for population heterogeneity and rational for potentially excluding mixed populations.

Validity of the findings

Conclusion and discussion that the finding of the MA contradict the NICE guidelines and that psychologic intervention initiated after ICU discharge in not needed is NOT supported by the results of this study. This is a potentially dangerous recommendation in fact. What is reported here is that the current approaches are not helping, and in the case of care givers making things worse, not that these patients don't need follow-up and intervention. They do have high levels of depression, anxiety, and PTSD and the current approaches are failing to address this. That is an important finding, but saying these patients don't need psychological follow-up and intervention is an incorrect and overreaching conclusion to what is reported in this meta-analysis. The authors findings in the current meta-analysis suggest that what was studied in the current sample of studies had no effect, not that monitoring and interaction is not necessary.

There are beginning to be recommendations regarding screening measures for all components of PICS, including mental health components. A recently published expert consensus statement regarding screening measures for PICS has been published. Spies CD, Krampe H, Paul N, et al. Instruments to measure outcomes of post-intensive care syndrome in outpatient care settings – Results of an expert consensus and feasibility field test. Journal of the Intensive Care Society. 2021/05/01 2020;22(2):159-174. doi:10.1177/1751143720923597. The US SCCM recommends The Impact of Event Scale – Revised (IES-R) for PTSD assessment. It is incorrect to state that there are no risk assessment models for psychologic dysfunction in survivors of critical care.

Annotated reviews are not available for download in order to protect the identity of reviewers who chose to remain anonymous.

---

## Round 0.2 · Minor Revisions

We appreciate the authors' efforts in response to the reviewer' comments, which have substantially improved the manuscript. There remain some comments which need addressing before a final decision on the publication of the paper can be made. In particular, reviewers pointed out some language-related issues. Please have the paper properly language edited by a professional service or a fluent speaker.

Reviewer 1 ·

Basic reporting

See below

Experimental design

See below

Validity of the findings

See below

Additional comments

Thank you very much for processing the feedback. The content of the article has improved a lot. I still have some small comments.

Regarding the experience of the researchers, this question refers to their team. What experience does your team have in terms of the method but also the clinical content?

When do they call patients:- patients, when critically ill patients?

You write...In our qualitative analysis, which included only RCTs more restrictively, our results contradicted the suggestions of previous guidelines and conclusions of a previous SR. This could be because of the differences in the target informal caregivers as well as different design used in the two SRs. What exactly do you mean by your "qualitative analysis"?

To what extent did memories of the ICU stay, possibly caused by psychological interventions, play a stressful role? - and these then lead to a worsening of symptoms?

Were nurses and doctors in the studies trained to deliver psychological interventions?

Reviewer 2 ·

Basic reporting

There are a few new additions (detailed below) that lack clarity and are confusing as written.

Introduction:

Line 95-97. Confusing as written. How can ICU follow-up occur prior to ICU discharge. ICU diaries and mobility are not “follow-up”, they are evidenced based interventions targeted at various PICS symptoms.

Line 99. Confusing as written. Unclear how psychological problems can have efficacy.

Materials & Methods:

Interventions. Line 155. Confusing as written. I am unclear about the timing of the psychologic interventions required “ after monitoring, counseling, and CBT.” As this is written it sounds like the intervention of choice are second or third line intervention after others have already been tried, which I do not think is the intention of the authors.

Data items:

Sensitivity analysis and subgroup analysis: Line 264-265. Confusing as written. As written it seems the authors conducted a subgroup analysis with caregivers, ICU survivors, and non-ICU survivors (which is obviously impossible). I believe the authors actually conducted a subgroup analysis with caregivers of both ICU survivors and non-survivors?

Results:

Study characteristics: Line 340. Same confusion regarding survivors as in subgroup analysis. I believe this study conducted interviews specifically with caregivers of non-survivors?

Discussion:

Line 507-510. Confusing who the authors are referring to in these two sentences as they group patients and caregivers into the first sentence and then start the second sentence the “they”. Who is “they” patients or caregivers, or both?

Experimental design

In the Materials & Methods section, specifically the population subsection, there is lack of clarity on the population of this SR & MA. The population in a SR & MA is the identified studies, not patients. Suggest that the “studies “and “information sources” paragraphs can be condensed and combined.

Validity of the findings

No comment

Additional comments

This reviewer appreciates the time and detailed effort the authors have spent in revisions. Overall, the additions and reviews strengthen the manuscript and it is much improved. There are still some minor grammatical/lack of clarity in writing issues to address. This is primarily in the newly added content.

---

## Round 0.3 · accepted · Accept

After positive feedback from reviewers, the paper is ready for publication. There is still a small error in Line 390. Incomplete sentence present. “if not necessary”. This should be corrected.

Reviewer 1 ·

Basic reporting

Dear authors.

Thank you for the careful editing of the manuscript. This editing has substantially improved the manuscript. I have no more comments.

Experimental design

no comment

Validity of the findings

no comment

Additional comments

Dear authors

Thank you for the careful editing of the manuscript. This editing has substantially improved the manuscript. I have no more comments.

Reviewer 2 ·

Basic reporting

Line 390. Incomplete sentence present. “if not necessary”. This should be corrected.

Experimental design

No comment

Validity of the findings

No comment

Additional comments

Revisions in this most current version are well done. One very minor grammatical/typographical error in line 390 that needs to be addressed.